# Reconstructing the post-glacial spread of the sand fly *Phlebotomus mascittii* Grassi, 1908 (Diptera: Psychodidae) in Europe

Edwin Kniha[1], Vít Dvořák[2], Stephan Koblmüller [3], Jorian Prudhomme[4,5], Vladimir Ivović[6], Ina Hoxha[1], Sandra Oerther[7,8,9], Anna Heitmann[10], Renke Lühken[10,11], Anne-Laure Bañuls[4], Denis Sereno [4,12], Alice Michelutti[13], Federica Toniolo[13], Pedro M. Alarcón-Elbal[14,15], Daniel Bravo-Barriga [16], Mikel A. González[17,18], Javier Lucientes[19], Vito Colella[20], Domenico Otranto[21,22], Marcos Antônio Bezerra-Santos [21], Gernot Kunz[3], Adelheid G. Obwaller[23], Jerome Depaquit[24], Amer Alić[25], Ozge Erisoz Kasap[26], Bulent Alten[26], Jasmin Omeragic[27], Petr Volf [2], Julia Walochnik [1], Viktor Sebestyén[28] & Attila J. Trájer [28✉]

Phlebotomine sand flies (Diptera: Phlebotominae) are the principal vectors of *Leishmania* spp. (Kinetoplastida: Trypanosomatidae). In Central Europe, *Phlebotomus mascittii* is the predominant species, but largely understudied. To better understand factors driving its current distribution, we infer patterns of genetic diversity by testing for signals of population expansion based on two mitochondrial genes and model current and past climate and habitat suitability for seven post-glacial maximum periods, taking 19 climatic variables into account. Consequently, we elucidate their connections by environmental-geographical network analysis. Most analyzed populations share a main haplotype tracing back to a single glacial maximum refuge area on the Mediterranean coasts of South France, which is supported by network analysis. The rapid range expansion of *Ph. mascittii* likely started in the early mid-Holocene epoch until today and its spread possibly followed two routes. The first one was through northern France to Germany and then Belgium, and the second across the Ligurian coast through present-day Slovenia to Austria, toward the northern Balkans. Here we present a combined approach to reveal glacial refugia and post-glacial spread of *Ph. mascittii* and observed discrepancies between the modelled and the current known distribution might reveal yet overlooked populations and potential further spread.

A full list of author affiliations appears at the end of the paper.

Phlebotomine sand flies (Diptera: Psychodidae: Phleboto-minae) are small hematophagous insects inhabiting tropi-cal, subtropical, and temperate regions. They are of significant medical and veterinary relevance as vectors of *Leishmania* spp. protozoans, bacteria, and several arboviruses capable of infecting humans in various regions of the Old and New Worlds[1]. Human leishmaniases are among the top ten neglected tropical diseases globally, with a burden of approximately 50,000 deaths per year and one billion people at risk of infection[2]. Leishmaniasis caused by *Leishmania infantum* occurs pre-dominantly in the tropics and subtropics, including the Medi-terranean area[3], and its incidence is expected to be underestimated in most countries in the WHO European Region[4]. This region, as well as Israel, Turkey, Turkmenistan, and Uzbekistan, are the most affected countries and account for almost 80% of the total number of cases reported in the region. Moreover, the Mediterranean North African Maghreb region is highly endemic for cutaneous leishmaniasis (CL), with Algeria being the second most affected country worldwide. Moreover, leishmaniasis also occurs in the Balkans as well as the southern Caucasus and Central Asia[2,5].

In Afro-Eurasia, various *Leishmania* species are transmitted to mammals by *Phlebotomus* sand flies. While the occurrence of sand flies and the endemicity of leishmaniasis in Mediterranean Europe have been known for decades, the occurrence of sand flies in the more continental territories (particularly Central Europe) has been less understood; however, recent findings record their emergence or sporadic and potentially overlooked presence[6]. In Central Europe, several sand fly species focus the attention of epidemiologists, namely *Phlebotomus* (*Larroussius*) *perniciosus* Newstead, 1911, *Ph.* (*L.*) *neglectus* Tonnoir, 1921, *Ph.* (*L.*) *perfiliewi* Parrot, 1930, *Ph.* (*L.*) *tobbi* Adler & Theodor, 1930, *Ph.* (*L.*) *ariasi* Tonnoir, 1921, and *Ph.* (*Phlebotomus*) *papatasi* (Scopoli, 1786) as vectors for *Leishmania* spp. and/or for phleboviruses[7–11]. Additionally, *Ph.* (*Transphlebotomus*) *mascittii* Grassi, 1908, and *Ph.* (*Adlerius*) *simici* Nitzulescu, 1931, are suspected but unproved vector species[12–15].

*Phlebotomus mascittii* has the widest and northernmost dis-tribution in Europe (Fig. 1). It has been reported from many Mediterranean countries of Europe, including the western Medi-terranean countries of Spain[16,17], France[18,19] including Corsica[20] and Italy[21–23], several Balkan countries of ex-Yugoslavia[24–28], and historically also from eastern Mediterranean countries of Greece and Turkey, even though these records shall be reviewed with respect to later descriptions of new species of the subgenus *Transphlebotomus* from this region[29]. The presence, however, is not restricted to the Mediterranean region and reaches into higher latitudes; the northernmost record is known from Germany[30]. Despite some recent records from Austria[31–33], Germany[34], Slovakia[35], and Hungary[36], its exact distribution in Central Europe is still largely unknown. The species distribution outside Europe is limited to a single finding in northern Algeria that represents its southernmost record[37].

Data about the capacity of sand fly species to adapt to envir-onmental changes allow us to predict changes in sand fly distribution[38]. For instance, previous studies allowed to increase scientific knowledge about the paleozoogeographical history of *Larroussius*[39], *Paraphlebotomus*[40–42] and *Phlebotomus*[43,44] sub-genera. Despite being one of the most widespread sand fly spe-cies in Europe, an accurate appraisal of the distribution and dispersal patterns of *Ph. mascittii* has yet to be conducted. It is known that glacial-interglacial climatic changes significantly impacted sand fly distributional ranges. During the last glacial period, which ended about 10,000 to 12,000 years ago, vast parts of Central and Eastern Europe were covered by permafrost and

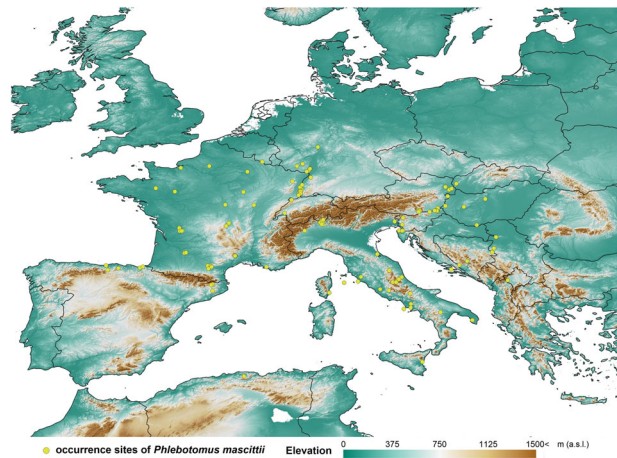

**Fig. 1 Described records of *Phlebotomus mascittii* used for modeling analyses.** Occurrence data in Sardinia and North Macedonia were not included in the analyses and were georeferenced to the center of the country due to a lack of point data. Molecularly analyzed specimens from this study are included in the map; details are given in Supplementary Data 1. Country border data is freely available at TM WORLD BORDERS 0.3 (URL: https://koordinates.com/layer/7354-tm-world-borders-03/) and elevation data at ETOPO Global Relief Model (URL: https://www.ncei.noaa. gov/products/etopo-global-relief-model).

tundra, resulting in a considerable decrease in pre-glacial bio-diversity and probably in the total disappearance of sand fly taxa from the continental regions of Europe[45]. Thus, being typical Mediterranean insects, sand flies either became extinct or sur-vived in (extra-) Mediterranean refugia[46,47]. The coldest periods drove sand flies and their pathogens to the southern Medi-terranean coastlines[45]. Due to the climate-ameliorating effect of the Mediterranean Sea, some sand fly populations also survived on the Mediterranean islands, which were colonized during the Messinian salinity crisis in the latest Miocene epoch by sand flies[29,39,41,48]. For example, genetic evidence suggests that some species, such as *Ph. ariasi* Tonnoir, 1921, had a refugium north of the Pyrenees during the Last Glacial Period[49].

Intensive surveys over the last decades have revealed that sand fly species occur in several, sometimes geographically distant regions in Europe, e.g., southern Belgium and northern Germany[18,30]. After the last glacial period, excluding the short but significant cooling of the Younger Dryas stadial[50], the onset of the Holocene was marked by a fast temperature rise and rapid transgression[51]. Two warm periods around 6500 and 4500 years ago (the Holocene optima) were characterized by mean tem-peratures similar to current climatic conditions. During these times, numerous animal and plant species from southern refugia colonized more northern regions, including Central Europe[47,52–54]. Also, sand flies (re-)colonized northern areas, established, and survived in small stable populations in micro-climatically favorable areas until today[45,47,55]. Species distribution models suggest that sand fly populations of different Mediterra-nean species could have occurred as far north as the British Isles during the Holocene optima[45]. The migration routes of sand flies to central parts of Europe are, however, complex and still poorly understood.

The present study aimed to elucidate the effect of climatic and sea level changes on the expansion of *Ph. mascittii* from its glacial refugia within the last 17 kiloyears (kys). Here, we used an integrated approach linking phylogeographic data, climate mod-eling, and network analysis to unlock new insights into the pro-cess of post-glacial sand fly colonization of Central Europe.

## Results

**Patterns of genetic diversity**. Altogether, 92 COI and 95 Cytb sequences of *Ph. mascittii* specimens were included in the genetic analyses (Supplementary Data 1). COI sequences without gaps and stop codons (Supplementary Data 2) with a length of 626 bp clustered into four haplotypes with three variable sites. The haplotype diversity (Hd) was 0.4190, and the nucleotide diversity (π) was 0.00074. One major haplotype (COI_2) was shared among 69 of 92 sequences from all included countries/territories except Corsica Island, with a distinct and unique haplotype (COI_4). While haplotype 1 (COI_1) consisted of specimens from Austria and Serbia, haplotype 3 (COI_3) only included sequences originating from Austrian representatives (Fig. 2a). The Cytb sequences with a total length of 641 bp also showed no gaps or stop codons (Supplementary Data 3). In total, nine haplotypes were identified, having eight variable sites, of which one was parsimony informative. The haplotype diversity (Hd) was 0.2708, and the nucleotide diversity (π) was 0.0006. Like in the COI haplotype network, one major haplotype was observed for Cytb, i.e., Cytb_1, including 81 of 95 the analyzed sequences, which originated from all included countries. Again, sequences of Corsican specimens clustered within a distinct haplotype (Cytb_3), except for one sequence that belonged to Cytb_1. Unlike the COI network, several singletons (Cytb_2, Cytb_4–Cytb_8) were observed in the Cytb network (Fig. 2b).

For COI haplotypes, a mean pairwise distance of 0.24% (range: 0.16–0.32%) was observed thereby (Supplementary Table 1), and Cytb haplotypes showed an overall mean pairwise distance of 0.3% (range: 0.16–0.49%) (Supplementary Table 2).

The star-shaped haplotype networks (Fig. 2), together with a significantly negative Tajima's D (D = −1.655; p = 0.018) and a unimodal mismatch distribution (Fig. 3a) with non-significant raggedness index (rg = 0.097; p = 0.711) but a marginally non-significant sum of squared differences (SSD = 0.004; p = 0.064), indicate a recent population expansion. Also, the Bayesian Skyline plot provided clear evidence for strong recent post-glacial population expansion (Fig. 3b). The time to the most recent common ancestor was estimated at 42.97 KY (95% highest posterior density interval 8.02–86.35 KY) to 107.42 KY (95% highest posterior density interval 20.05–215.87 KY), depending on the assumed substitution rate. No evidence for isolation by distance was present in our data (Supplementary Fig. 1, Supplementary Data 4).

**Modeled alterations of climatic suitability patterns and former ranges**. The model shows that areas with high climatic suitability in Heinrich Stadial 1 existed along the currently partly submerged coasts of the Gulf of Lyon, as well as on the narrow coastline of the Ligurian Sea, as well as in the south-central regions of the Apennine Peninsula. Compared to the previous period, the high-suitability regions of the Bølling–Allerød Interstadial show a larger extension in present-day France, including territories adjacent to the Bay of Biscay and the Gulf of Lyon. However, for the Younger Dryas Stadial, the suitability patterns could be like those in the Heinrich Stadial 1 period. Although in the Greenlandian (early Holocene) period, the climatically suitable areas exhibit a more minor increase compared to the late Pleistocene conditions. From the Northgrippian (mid-Holocene) period onwards, the extension of the climatically suitable regions began to expand rapidly, covering large areas of continental western Europe and the Apennine Peninsula. This rapid expansion in regions with high climatic suitability was continuous between the Northgrippian and the Anthropocene and led to the colonization of Central Europe (Fig. 4).

The model results indicate that the occurrence of *Ph. mascittii* in the late Pleistocene era could have been restricted to relatively small regions of the Mediterranean coasts of Southwest Europe and the Apennine Peninsula. Between 17 and 11.7 ka, the patterns of the potential range of the species increased, but the significant expansion of the species only started after the Greenlandian period, in the early mid-Holocene era. Parallel to this areal expansion, the species could have lost a notable area of their glacial refugia due to the post-glacial transgression. Based on the modeled ranges, a relatively dispersed Southwest European along the Mediterranean coasts and an Apennine Peninsula glacial refugia can be hypothesized. Among the sampling sites of this study, only the sites in Southwest France can be found within the Southwest European glacial refugium of the species (Fig. 5).

**Network analysis results**. Considering the percental frequency (%) of the number of periods with climatic suitability for *Ph. mascittii*, when the local climatic suitability overwhelmed 90% in a site, the Mediterranean coasts of South France could have been a stable refugia over the past 17 kys. The geographical trend of the climatic suitability values suggests that the post-glacial migration of *Ph. mascittii* had an eastward character. The relative centrality values show that the Gulf of Lyon and the Swiss Middle Mountain region could have played a key role in the eastward migration of the species in the post-glacial era (Fig. 6). The network analysis results indicate that the peripheral region of this refugium likely played the most significant role as a source of the spread of the species, while Southwest France represents the climatically most stable glacial refugial core region (Fig. 7). The result of network similarity hierarchical clustering denotes a strong connection between the two genes (r² = 0.31) and between climatic and spatial factors (r² = 0.21). A weak correlation was observed between genetic and climatic as well as spatial factors, and the

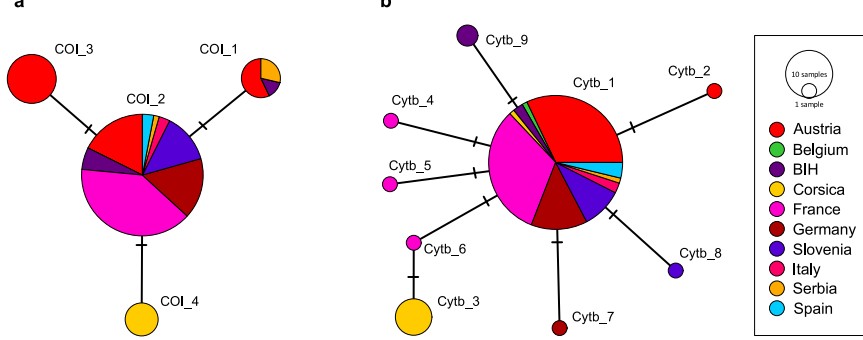

**Fig. 2 Statistical parsimony networks of *Phlebotomus mascittii* from eight countries.** Analysis based on COI (**a**) and Cytb (**b**) sequences. BIH Bosnia and Herzegovina.

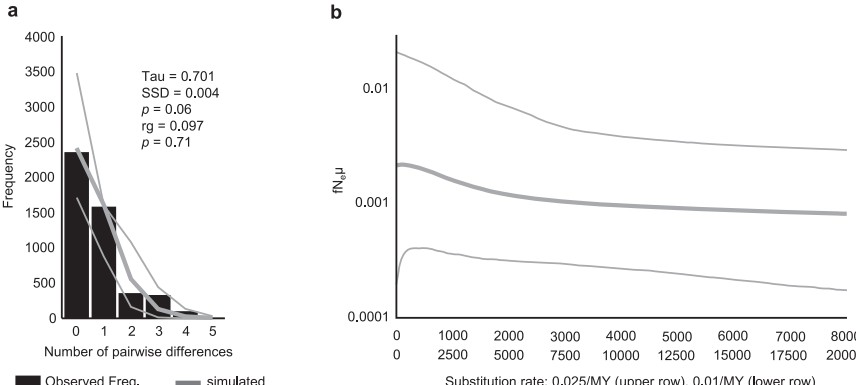

**Fig. 3 Signatures of population expansion in *Phlebotomus mascittii*. a** Mismatch distribution for *Ph. mascittii* in Europe, based on combined COI and Cytb sequences. Black columns represent the observed frequency of pairwise differences. Gray lines refer to the expected distribution based on parameter estimates and their 95% confidence limits simulated under a model of population growth. The sum of squared differences (SSD) and raggedness index (rg) and their respective *P* values are given to describe the fit of the observed mismatch distribution to the expectation based on growth parameter estimates. **b** Bayesian Skyline Plot (BSP), assuming a minimum and maximum substitution rate for mitochondrial protein-coding genes of insects of 1% and 2.5% per. The thick line denotes the median estimate; thin lines indicate the 95% highest posterior density (HPD) interval. $fN_e\mu$ = female effective population size scaled by substitution rate. Raw data of the analyses are given in Supplementary Data 2.

phylogeographic patterns were not supported by the sites' general climatic suitability or the points' distance (Supplementary Fig. 2).

## Discussion

Using an integrated approach that encompasses phylogeographic data, climate modeling, and network analysis, we unlocked insights into the post-glacial expansion of *Ph. mascittii* in Europe and investigated its refugia and colonization routes. The molecular results suggest that *Ph. mascittii* populations experienced a notable genetic bottleneck before their post-glacial expansion from a southwestern European refugium. A major glacial refugium was identified in southern France by comparing the genetic data and modeled former climatic patterns. Another potential Apennines refugial area might be assumed but needs further clarification with currently unavailable data. The rapid post-glacial expansion of *Ph. mascittii* started in the early mid-Holocene era, and it might have reached its current range during the Meghalayan stage. The Southwest European region north of the Pyrenees likely played a key role in the start of the post-glacial migration of the species; however, the survival of *Ph. mascittii* during cold periods was due to the Southern France refugium. A partly similar region is assumed to have served as the most important glacial refugium for *Ph. ariasi*[49].

Notably, the currently submerged epicontinental shelf areas, which were drylands during the Last Glacial Maximum, could have played an important role in the survival of sand fly species. This fact is most visible, e.g., in the case of the Gulf of Venice in the North Adriatic region. Despite little knowledge of the glacial paleoenvironmental conditions of these regions, some authors suggest that *Quercus* species, which formed deciduous forests, returned to West and Central Europe in the postglacial era[56]. However, recent investigations showed that temperate seasonal forest habitats were widely present in the peri-Mediterranean area even in the late glacial era[57]. Currently, *Ph. mascittii* is known to occur in deciduous forests in Central Europe, e.g., in the Kapolcs Valley, Hungary[58], in close vicinity to a deciduous forest dominated by *Quercus cerris* L. Trájer[57] showed that the southern regions of France during the Bølling-Allerød period could have supported temperate seasonal forests, which would have been a suitable habitat for *Ph. mascittii*. This is noteworthy, considering that the (now submerged) coastal plains could have provided an tempered and relatively humid environment for *Ph. mascittii* populations—environmental conditions in which this sand fly

species thrives[59]. We, therefore, hypothesize that *Ph. mascittii* populations could have survived the Last Glacial Maximum cold in their South European refugia in deciduous forest habitats, and their postglacial expansion was linked to the northward spread of the temperate seasonal forests.

The recent post-glacial expansion of *Ph. mascittii* is well supported by the observed star-shaped haplotype networks and the unimodal mismatch distribution. Similar processes have been hypothesized for *Ph. papatasi*[60] and *Ph. perniciosus*[61]. Also, the inferred Skyline Plot shows a recent population expansion, particularly in the last 10,000 to 5000 years, which aligns well with our climate modeling results. Similar to the post-glacial migration patterns of *Ph. perniciosus*[62], we did not observe evidence of genetic isolation by distance due to the short analyzed timespan. The weak correlation between the two genetic, climatic, and spatial factors might be best explained by the lack of recombination in the mitochondrial genome and the linkage between the two analyzed loci. Hence, the low correlation between the two loci in this dataset has a stochastic effect, namely a very recently analyzed time scale in which only a few substitutions have occurred. In other possible cases, e.g., with the multiple previous refugia, at least the spatial factors would show a moderate or more substantial correlation with the genetic traits. We assume that small populations of sand flies migrated to the east after the post-glacial era from the Southwest Ice Age refugium in several waves during the last 6 kys, as also demonstrated for other insect taxa, e.g., grasshoppers (Orthoptera)[63,64].

Based on our combined phylogeographic and modeling approaches, we hypothesize two potential, possibly parallel, dispersal routes from a southwestern European refugium: firstly, through the Côte d'Azur, the Po Valley, and the Slovenian mountain passes; and/or secondly, a northern route via Northeast France and South Germany. Geographical and climatic conditions must be considered regarding potential post-glacial migration routes, starting in the Pyrenees region. After the mid-Holocene warming, the main barriers to the west-east spread were the Alps in Central Europe, the Apennines in the Mediterranean area, and the Dinaric mountains. According to the model results, *Ph. mascittii* populations could have avoided the Alps by the northern route and reached the Carpathian Basin. Considering the southern way, crossing the northern Dinaric areas could have occurred via passes at lower altitudes. However, anthropogenic factors must be considered. Human farmer populations reached mainland Europe

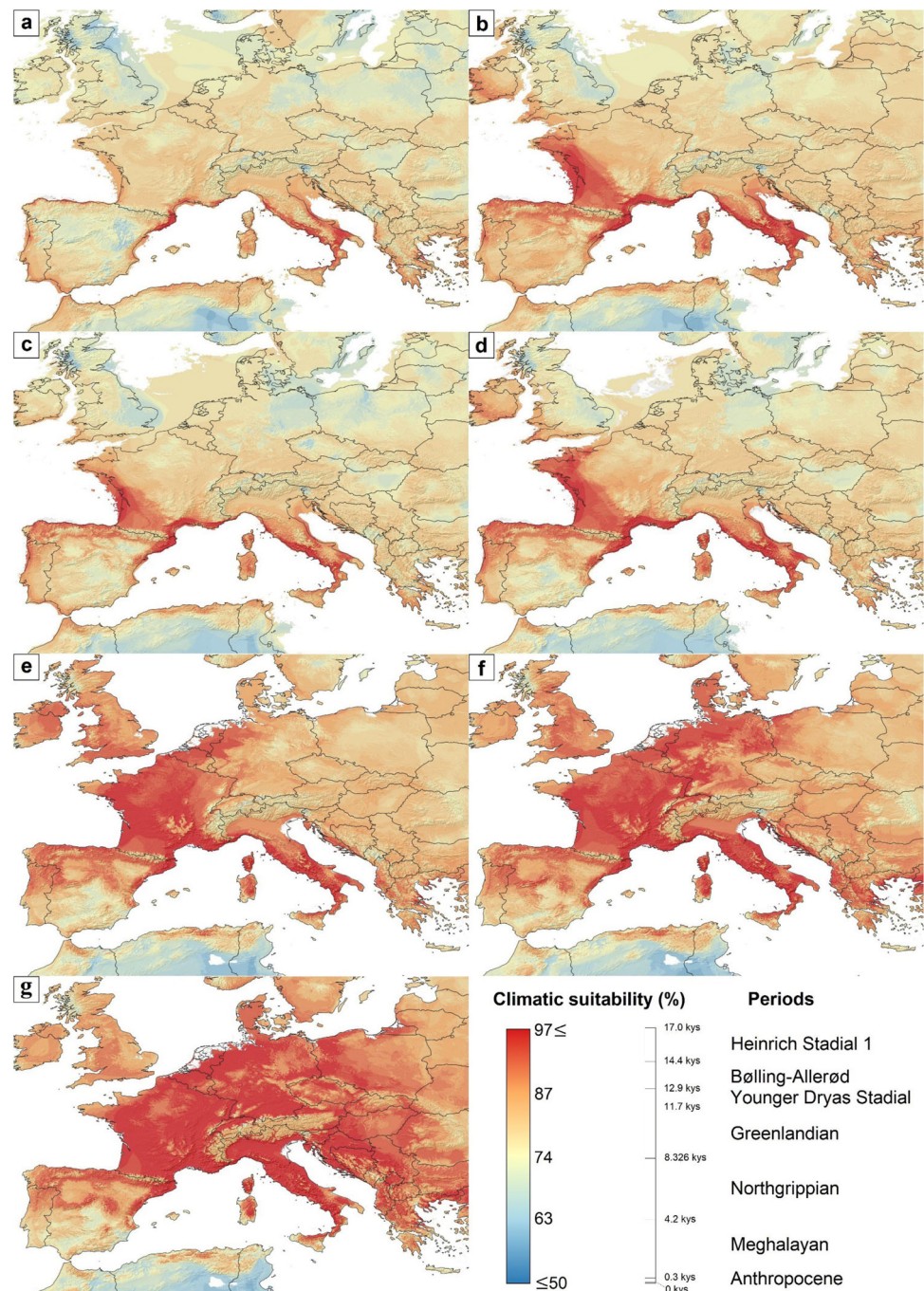

**Fig. 4 The changes in climatic suitability for *Phlebotomus mascittii* over the last 19 kys. a** Heinrich Stadial 1 (17.0–14.7 kys), **b** Bølling-Allerød (14.7–12.9 kys), **c** Younger Dryas Stadial (12.9–11.7 kys), **d** Greenlandian (11.7–8.326 kys), **e** Northgrippian (8.326–4.2 kys), **f** Meghalayan (4.2–0.3 kys), and (**g**) Anthropocene (1979–2013). Modeling results were georeferenced in paleoclimatic maps, openly available at PaleoClim.org (URL: http://www.paleoclim.org/) and WorldClim (URL: https://www.worldclim.org/).

roughly in 6700 Before Christian Era (BCE), the broader environment of the Pyrenees in 5800 BCE, and from 5200 BCE onwards, both the northern and southern forelands of the Alps and the Dinaric mountains were populated by Neolithic societies[65]. Overall, the abovementioned factors may have contributed to the colonization and spread of *Ph. mascittii* in Europe. Noteworthy, the currently observed distribution shows marked discrepancies from the current modeled climatic suitability of this species. This might be explained by the fact that *Ph. mascittii* has been reported to favor urban and semi-urban breeding sites such

as animal farms, old barns with natural floors or even cemeteries and is, thus, mostly restricted to these sites[16,33].

To uncover additional potential migration routes, it will be necessary to carefully study currently unavailable specimens from Italy and the northern Balkans to understand the role of potential refugia in the Apennine Peninsula and the Balkans. The hypothesized southwest-to-northeast or south-to-north direction of post-glacial migration trends is in agreement with observed diversity patterns in other cold-sensitive animal groups like reptiles and amphibians[66].

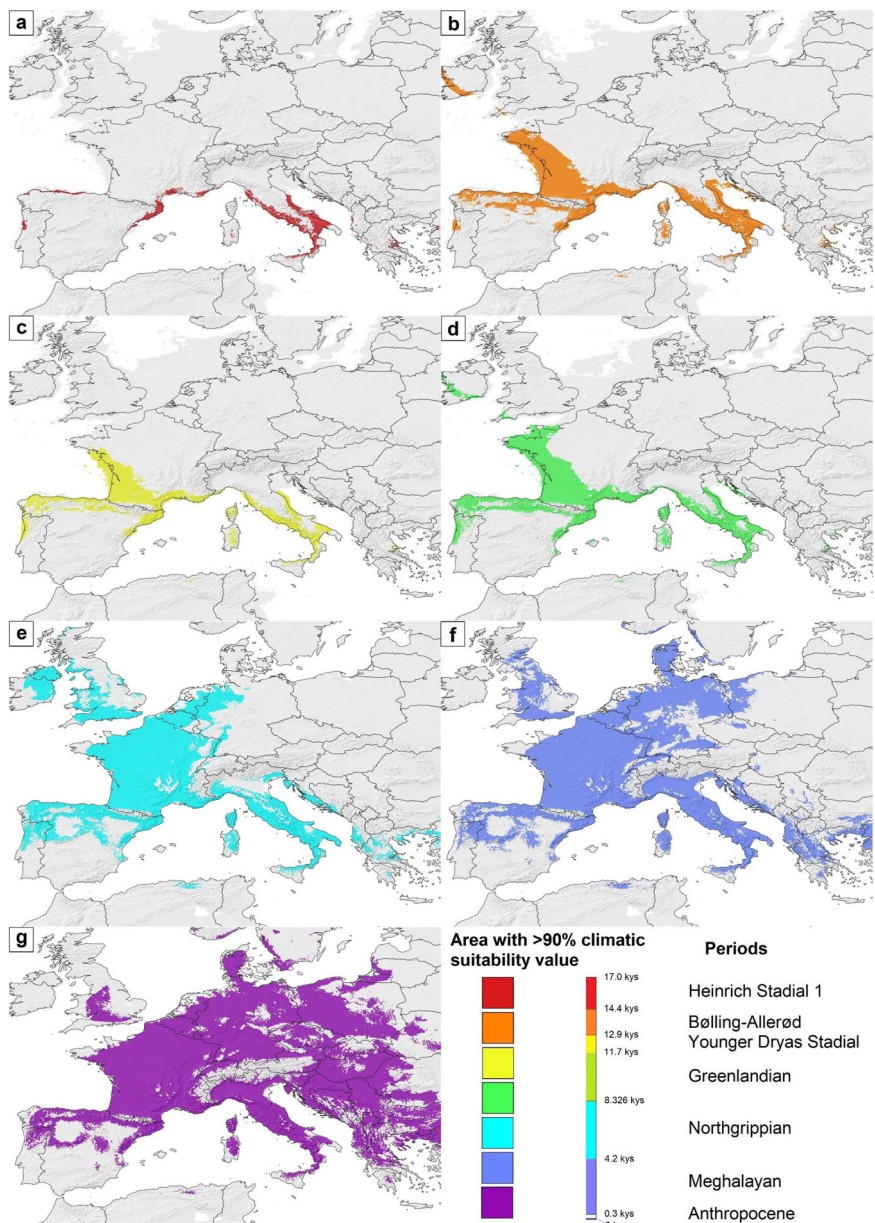

**Fig. 5 The changing expansiveness of areas with more than 90% climatic suitability values for *Phlebotomus mascittii* in the last 17 kys. a** Heinrich Stadial 1 (17.0–14.7 kys), **b** Bølling-Allerød (14.7–12.9 kys), **c** Younger Dryas Stadial (12.9–11.7 kys), **d** Greenlandian (11.7–8.326 kys), **e** Northgrippian (8.326–4.2 kys), **f** Meghalayan (4.2–0.3 kys), and (**g**) Anthropocene (1979–2013). Modeling results were georeferenced in paleoclimatic maps, openly available at PaleoClim.org (URL: http://www.paleoclim.org/) and WorldClim (URL: https://www.worldclim.org/).

The results herein presented also indicate a post-glacial dispersal of *Ph. mascittii* to Corsica. The present-day distance between the nearest dryland points of Corsica and the Apennine Peninsula is about 81 km, and between the islands of Corsica and Elba is approximately 50 km. Although these distances were shorter in the Last Glacial Maximum due to the low sea level, this still means a considerable distance for sand flies due to their frequently publicized weak flying capability. For example, the maximal travel distances recorded for *Ph. papatasi* females and males were 1.91 and 1.51 km, respectively[67]. However, Tonelli et al.[68] showed that different sand fly species may show different flight abilities and might reach greater distances than previously reported. Thus, this mode of natural dispersal, particularly involving wind, should at least be considered, although never observed for sand flies and controversially discussed. Sand flies are tiny insects, very sensitive to low air humidity (and the

consequent desiccation), large temperature variations, and intensive solar radiation[69]. Thus, since these conditions prevail during transport by wind over the sea, it remains uncertain if these insects could have survived such a dispersal mode. However, the dispersal from the mainland to Corsica could also have been facilitated by human activities. Humans settled in Corsica around 6000 BCE[70] and the seafaring population brought sheep, goats, and pigs to the island[71], which implicates the transport of litter as well as bulks of sand or soil below deck. This mode of dispersal has also been discussed for the island of Santorini, Greece[72], and for *Sergentomyia clydei* (Sinton, 1928) to the Seychelles, and *Sergentomyia babu* (Annandale, 1910) to Mauritius[73].

This study provides to the best of our knowledge the first detailed insight into the most likely post-glacial dispersal of *Ph. mascittii*, the currently most widely distributed sand fly species in Europe. Our approach, combining phylogeographic, climatic, and

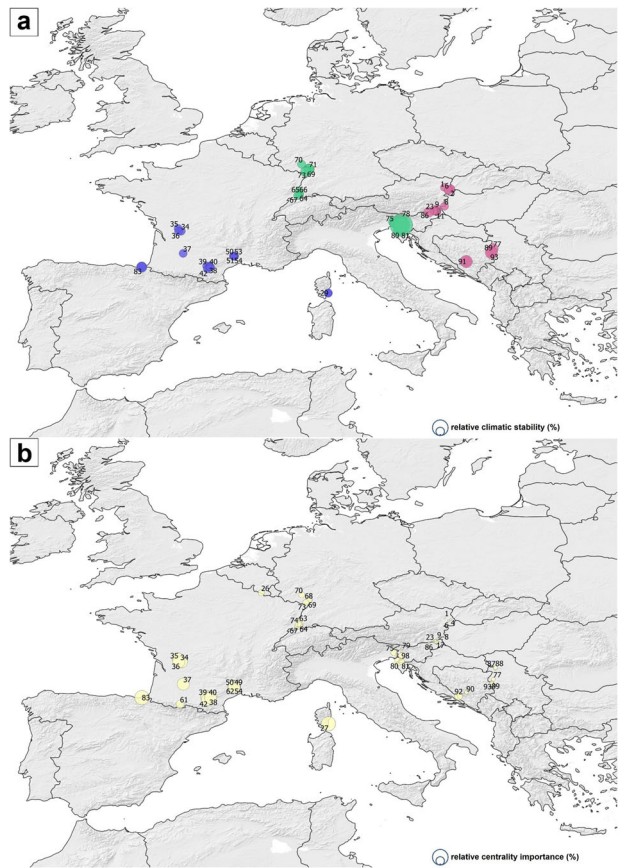

**Fig. 6 Relative climatic and relative centrality measures of the sampled *Phlebotomus mascittii* sites.** The diameter of the circles represents the relative climatic suitability values (%) (**a**) and centrality importance (%) (**b**). The numbers are in accordance with the site numbers sorted in Supplementary Data 1. Network results were georeferenced in paleoclimatic maps, openly available at PaleoClim.org (URL: http://www.paleoclim.org/) and WorldClim (URL: https://www.worldclim.org/).

networking data, revealed a crucial glacial refugia in southwestern France. Based on the results of this study, we hypothesize a recent post-glacial dispersal of *Ph. mascittii* by two different routes. Elucidating post-glacial sand fly migration patterns to Central Europe might help to predict future sand fly dispersal and that of pathogens potentially transmitted by them.

## Material and methods

**Sampling and morphological sand fly identification.** For this study, 38 complete specimens and 48 DNA samples of *Ph. mascittii* obtained during various entomological surveys, including published and yet unpublished material, were used. In addition, the dataset was complemented with 14 previously published *Ph. mascittii* sequences available in the GenBank database. The head and terminal segments of the abdomen of all unidentified specimens were dissected and slide-mounted on a glass slide in CMCP-10 high-viscosity mountant (Polysciences Europe GmbH, Hirschberg an der Bergstraße, Germany). Identification was based on the morphological parameters of the male genitalia, female spermatheca, and pharyngeal armature[74].

**DNA extraction, PCR amplification and sequencing.** The remaining body parts of slide-mounted specimens were transferred into individual vials (1.5 mL), 180 µL ATL buffer and 20 µL Proteinase K were added, and homogenization was performed with a hand homogenizer (Thomas Scientific, Svedesboro, NJ, USA).

DNA isolation was performed with a QIAamp® DNA Mini Kit 250 (Qiagen, Hilden, Germany), following the manufacturer's protocol. DNA was eluted in a final volume of 100–200 µL of elution buffer. Sand flies from Germany were homogenized in 180 µL of Dulbecco's Modified Eagle Medium (Sigma Aldrich, St. Louis, Missouri, United States (US)), 100 µL homogenate was used for DNA extraction using the 5x MagMax Pathogen RNA/DNA Kit (Thermo Fisher Scientific, Waltham, Massachusetts, US), DNA was eluted in 80 µL. Partial sequences of two mitochondrial genes were amplified by PCR, namely cytochrome c oxidase subunit 1 (COI) and cytochrome b (Cytb). For the COI, a 658-bp fragment was amplified using the primers LCO1490 (5′-GGTCAACAAATCATAAAGATATTGG-3') and HCO2198 (5′-TAAACTTCAGGGTGACCAAAAAATCA-3')[75] and for Cytb, a 737-bp fragment was amplified using the primers CB1-SE (5′-TATGTACTACCCTGAGGACAAATATC-3') and CB-R06 (5′-TATCTAATGGTTTCAAAACAATTGC-3')[76]. All PCR amplifications were run with the following PCR conditions: 95 °C for 15 min, followed by 35 cycles of 95 °C for 1 min (denaturation), 52 °C for 1:30 min (annealing), and 72 °C for 2 min (elongation), followed by a final extension of 72 °C for 10 min. PCR amplification was performed in reaction volumes of 50 µL containing 10× reaction buffer B, 2.5 mM MgCl₂, 1.6 mM dNTPs, 1 µM primers, 1.25 units DNA polymerase, and 1–5 µL DNA; sterile H₂O was added to the final volume of 50 µL. Thermal cycling was performed with an Eppendorf Mastercycler modular PCR system (Eppendorf AG, TM, Hamburg, Germany).

Bands were visualized and analyzed with a Gel Doc™ XR+ Imager (Bio-Rad Laboratories, Inc., Hercules, CA, USA), cut out of the gel, and purified with an Illustra™ GFX™ PCR DNA and Gel Purification kit (GE Healthcare, Buckinghamshire, UK). Sanger sequencing was performed using a BigDye® Terminator v.1.1 Cycle Sequencing kit (Thermo Fisher Scientific Inc, Waltham, MA, USA) and run on a SeqStudio® Genetic Analyzer (Thermo Fisher Scientific Inc., Waltham, MA, USA).

**Sequence editing and alignment.** Overall, 86 new COI sequences with a length of 658 bp and 87 new Cytb sequences with a length of 748 bp of individual *Ph. mascittii* specimens were obtained from both strands, and aligned and consensus sequences were generated using the DNA sequence analysis tool GeneDoc 2.7.0[77]. DNA chromatogram files were checked for double signals, if necessary. Translation to amino acid sequences of coding regions (COI and Cytb) showed intact reading frames for all included sequences, and no internal stop codons were observed (Supplementary Data 3, 4). Sequence identities were checked by comparing the obtained sequences to available sequences in GenBank using BLAST. Sequences were submitted to GenBank and are available under the following accession numbers: OQ064321.1–OQ064395.1 and OR573677.1–OR573687.1 for COI and OQ067398.1–OQ067475.1 and OR574850.1–OR574858.1 for Cytb (Supplementary Data 1). In addition, six COI sequences and eight Cytb sequences of *Ph. mascittii* were downloaded from GenBank and included in the analysis (Supplementary Data 1). For each locus, sequences were aligned using ClustalX[78].

**Genetic diversity through space and time.** Unique haplotypes, haplotype (Hd), and nucleotide (π) diversity per locus were calculated with DnaSP v.5[79], and uncorrected pairwise distances between all haplotypes per locus were calculated in MEGAX[80]. Phylogenetic relationships among haplotypes were visualized as a statistical parsimony network[81], as inferred in PopART[82]. To test for signals of population expansion, we calculated Tajima's D[83] (1000 simulated samples) and a mismatch distribution (1000 bootstrap replicates) in Arlequin v3.11[84] based on the

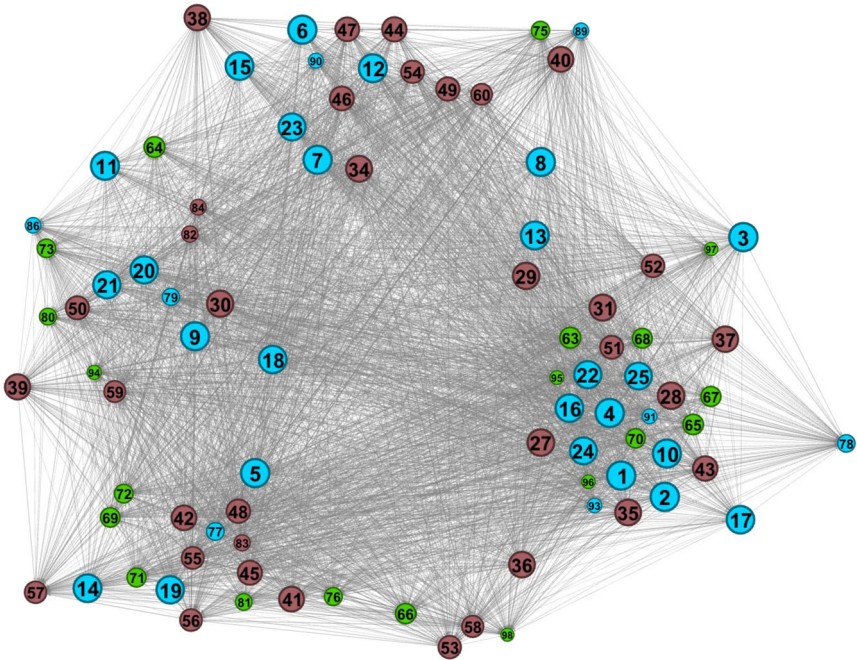

**Fig. 7 Results of the network analysis of the sampled sites.** Analysis based on the geographical, cytochrome c oxidase subunit I (COI) and cytochrome b (Cytb) sequence distances and the climatic suitability differences of the *Phlebotomus mascittii* sites. The size of the nodes represents their relative importance in the network based on the analyzed aspects of the sites. The numbers follow the site numbers given in Supplementary Data 1.

concatenated dataset, including both loci. The fit between the observed mismatch distribution and the expectations based on growth parameter estimates was evaluated by the sum of squared differences (SSD) and the raggedness index (rg). Furthermore, past population size trajectories and time to the most recent common ancestor (tMRCA) were inferred using a Bayesian coalescent approach in BEAST 2.7.1[85]. Due to the very low levels of variation observed within each locus, we refrained from partitioning the data per gene or codon position. We employed the best-fitting model of molecular evolution[86] selected by the Bayesian Information Criterion in ModelFinder[87], a strict molecular clock with a minimum and maximum substitution rate of 0.01 and 0.025 substitutions per site per million years (MY)[8,88], which covers the range of substitution rates typically inferred from / employed for mitochondrial protein coding genes in insects[60,89–91], and the Bayesian Skyline tree prior. Four independent MCMC runs of ten million generations each were done, sampling every 1000th step and a burn-in of the first 10% of sampled generations. LogCombiner (part of the BEAST package) was used to combine the individual log and tree files. Assessment of run convergence, verification of effective sample sizes [ESS > 200 for all parameters, indicating that the parameter log file accurately reflects the posterior distribution[92]], and visualization of past population size changes were done in Tracer 1.7[93]. The Mantel test, as implemented in Alleles In Space 1.0[94], was used to test for a correlation between the genetic and geographical distances of the sampled individuals.

**Climatic and topographic data sources**. Four Holocene and three late Pleistocene models were applied in the modeling process. As a reference (current Anthropocene: 1979–2013) period, the model of Karger et al.[95] was used. Three other Holocene periods, the Greenlandian, Northgrippian, and Meghalayan periods, were also used. The Heinrich Stadial 1, Bølling-Allerød interstadial, and Younger Dryas Stadial periods' models represented the late Pleistocene (Table 1).

**Table 1 The sources of the used climatic models and data. The values of the Anthropocene climate model were used for the derivation of the range-limiting climatic extrema of sand flies.**

| Model period | Age | Reference |
|---|---|---|
| Current period (Anthropocene) | 1979–2013 | Karger et al.[95] |
| late-Holocene, Meghalayan | 4.2–0.3 ka | Fordham et al.[105] |
| mid-Holocene, Northgrippian | 8.326–4.2 ka | Fordham et al.[105] |
| early-Holocene, Greenlandian | 11.7–8.326 ka | Fordham et al.[105] |
| Younger Dryas Stadial | 12.9–11.7 ka | Fordham et al.[105] |
| Bølling-Allerød | 14.7–12.9 ka | Fordham et al.[105] |
| Heinrich Stadial 1 | 17.0–14.7 ka | Fordham et al.[105] |

**Geographical data and mapping**. A literature search for records of *Ph. mascittii* species was performed (Google Scholar and PubMed), and coordinates or locations were extracted. Published as well as yet unpublished *Ph. mascittii* trapping sites were georeferenced into a distribution map using Quantum GIS 3.4.11[96] (Fig. 1). For most of the occurrences, the exact coordinates were known (coordinates were known for all molecularly analyzed specimens). In the remaining cases, the location of the original trappings has been defined as accurately as possible. In some cases, the exact coordinates of the catch sites could be reconstructed. By identifying the geographical position of the streets and individual houses within the small settlements, the coordinate extraction was possible within a resolution accuracy of ten meters. In the case of other small settlements where a schematic map was not provided, the center coordinates of the villages were considered in georeferencing. Sometimes, the resolution was between $1 \times 1 <$ to $\le 10 \times 10$ km. Coordinate accuracy was ensured by a general resolution that ranged from $1 \times 1 <$ to $\le 10 \times 10$ km. Of all occurrence data points, 102 were exact point coordinates, 44 were at a scale of $1 \times 1$ km or lower, and eleven were at $10 \times 10$ km or lower. From the point of view of distribution modeling, even the accuracy of $\le 10 \times 10$ km resolution

is generally acceptable because these model experiments aim to predict large-scale occurrence patterns and not local habitats. Furthermore, a cut of 2–2 percentiles from the sampled climatic data was utilized to filter the possibly existing non-relevant climatic values. Finally, it is worth mentioning that the resolution of the used paleoclimate models is 2.5 arcminutes (in the mid-latitudes, it is equal to ~5 km) which means that the accuracy of the site coordinates fits the resolution of the climatic models. Supplementary Data 1 shows the essential data relevant to the studied sites.

**Modeling of climatic suitability patterns.** For the acquisition of the range-limiting extrema of *Ph. mascittii*, the 2.5 arcmin resolution model of Karger et al. (2021) related to the reference period of 1979–2013 was used. A total of 19 bioclimatic variables[97] were utilized among which 11 (bio1–11) have temperature and 8 (bio12–19) have precipitation nature (Supplementary Table 3). To gain data on distribution-limiting climatic extrema, 2-2 percentiles were cut from the absolute maximum and minimum values of the factors to avoid the involvement of unrealistic climatic contracts according to the generally applied considerations of environmental modeling[98]. The climatic values used in modeling are available in Supplementary Table 3.

The modeling of the former distribution areas followed the logic of Boolean algebra[99]. This technique is based on climate envelope or multidimensional modeling[100], which considers independently of the effects of environmental factors on the occurrence probability of species. Then, it estimates distribution areas or species-related climatic suitability values regarding its recorded occurrences. Using the climatic hypervolume contained by the minimum and maximum values of climatic variables within the native range of the researched species, a Boolean map of risk zones is produced in this procedure. The theoretical basis of this modeling technique is fundamentally based on the barrel analogy of Liebig's law of the minimum[101].

To generate the results, the raster calculator function of Quantum GIS 3.4.11[96] was used. Supplementary Data 5 shows the script of the utilized equation.

The area of all range areas was modeled according to the following general equations:

$$1(v_n) = \begin{cases} 0 & if \quad v_{n\_limit\_min} > v_n \ and \ v_n > v_{n\_limit\_max} \\ 1 & if \quad v_{n\_limit\_min} \le v_n \ and \ v_n \le v_{n\_limit\_max} \end{cases}$$

Where $v_n$ represents the $n$th climatic constraint of the distribution area of a species, $v_{n\_limit\_min}$ and $v_{n\_limit\_max}$ are the lower and upper distribution-limiting values related to the climatic constraint (Supplementary Table 4).

The potential area-based suitability patterns were determined according to the following mathematical formalism:

$$A(v_1; v_2; \dots; v_n) = 1(v_1) - v_1 \bigcap 1(v_2) - v_2 \bigcap \dots \bigcap 1(v_n) - v_n$$

where $A(v_1; v2 \dots v_n)$ shows the potential distribution area of the given species, which contains the remaining areas after considering the factor-related limitations.

Then, the modeled values were transformed into percentage (%) values and colorized.

**Network-based evaluation of the spatial similarities of modeled climatic suitability and genetic patterns.** The sampling points are considered a network, where $N = 88$ represents the nodes connected in an $A = N \ x \ N$ adjacency matrix. The $A$ adjacency matrix is symmetric due to the undirectedness of the defined network. The edge from node $j$ to node $i$ can be expressed as $e_{ij}$. Networks are defined for each examined variable, so the following

multilayer $M$ network can be written[102]:

$$M = \left( COI, Cytb, d, \sum_{a=1}^{7} S_a \right)$$

where $M$ represents the multilayer network, COI and Cytb stand for the identified genetic haplotypes, $d$ denotes the coordinates, and $S_a$ refers to the modeled climatic suitability values in the given $a$ model period. Points with missing genetic information ($n = 10$) were excluded from the network analysis (Supplementary Data 1). Network analysis was performed in Gephi 0.10.1[103].

The genetic distance values are given for all ($e_{ij}$) edges, and the geographic distance $(d)$ can be calculated from the coordinates. The climatic suitability values can be obtained from the difference in the cumulated $S_a$ value of the two nodes, so it is used to express a degree of climatic similarity. Finally, integrating the effects of the variables, an edge with multiple weights is introduced as follows:

$$w_{e_{ij}} = dCOI_{e_{ij}} + dCytb_{e_{ij}} + d_{e_{ij}} + S_{a_{e_{ij}}}$$

where the raw edge weights are normalized and their reciprocal value is used. Therefore, an edge with a high weight represents genetic similarity, relatively close nodes, and similar climatic conditions.

Based on the multi-weighted edges, the centrality measures of the identified network nodes are used to compare genetic, spatial, and climatic factors and to express the similarity of the patterns of the edges based on the Jaccard index[104].

**Reporting summary.** Further information on research design is available in the Nature Research Reporting Summary linked to this article.

## Data availability

All the data supporting the findings of this study are available within the article and its Supplementary Information files. All DNA sequence data was uploaded to GenBank under the accession numbers OQ064321.1–OQ064395.1, OR573677.1–OR573687.1, OQ067398.1–OQ067475.1, and OR574850.1–OR574858.1, and accession numbers are additionally given in Supplementary Data 1. Source data for charts and graphs are provided in Supplementary Data 2.

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

## Acknowledgements

The research presented in this article was Prepared in the "Climate Change Multi-disciplinary National Laboratory" project (RRF-2.3.1-21-2022-00014), within the framework of the Széchenyi Plan Plusz program, with the support of the Recovery and Resilience Instrument of the European Union. Data collection and analysis was supported by the Austrian Agency for Education and Internationalization (OeAD) [grant number WTZ CZ02/2020], the VectorNet project, a European network for sharing data on the geographic distribution of arthropod vectors, transmitting human and animal disease agents (Contract OC/EFSA/AHAW/2013/02-FWC1) funded by the European Food Safety Authority (EFSA) and the European Center for Disease Prevention and Control (ECDC), and by the Austrian defense research program FORTE of the Federal Ministry of Finance (BMF) [grant number 886318]. R.L. is funded by the Federal Ministry of Education and Research of Germany (grant number 01Kl2022). The authors are also deeply thankful to Iveta Häfeli for technical support and to Univ.-Prof. Dr. Horst Aspöck for substantial discussions, both members of the Institute of Specific Prophylaxis and Tropical Medicine, Medical University of Vienna.

## Author contributions

Design of the study: E.K., V.D., J.W., V.S. and A.J.T. Entomological fieldwork: E.K., V.D., J.P., V.I., S.O., A.L.B., D.S., A.M., F.T., P.M.A.E., D.B.B., M.A.G., J.L., V.C., D.O., M.B.S., G.K., A.G.O., P.V., A.A., O.E.K., B.A. and J.O. Laboratory work: E.K., V.D., V.I., I.H., A.H., R.L., A.M., F.T. and O.E.K. Data analyses: E.K., V.D., S.K., J.D., J.W., V.S. and A.J.T. Drafting of manuscript: E.K., J.W., V.S. and A.J.T. Critical review of manuscript: all authors.

## Competing interests

The authors declare no competing interests.

## Additional information

[1]Institute of Specific Prophylaxis and Tropical Medicine, Center for Pathophysiology, Infectiology and Immunology, Medical University of Vienna, Vienna, Austria. [2]Department of Parasitology, Faculty of Science, Charles University, Prague, Czech Republic. [3]Institute of Biology, University of Graz, Graz, Austria. [4]UMR MIVEGEC (Université de Montpellier—IRD—CNRS), Institute of Research for Development, Montpellier, France. [5]INTHERES, Université de Toulouse, INRAE, ENVT, Toulouse, France. [6]Department of Biodiversity, FAMNIT, University of Primorska, Koper-Capodistria, Slovenia. [7]Institute of Global Health, Heidelberg University, Heidelberg, Germany. [8]German Mosquito Control Association (KABS), Speyer, Germany. [9]Institute for Dipterology (IfD), Speyer, Germany. [10]Department of Arbovirology, Bernhard Nocht Institute for Tropical Medicine, Hamburg, Germany. [11]Research Group Vector Control, Bernhard Nocht Institute for Tropical Medicine, Hamburg, Germany. [12]Institut de Recherche pour le Développement, Université de Montpellier, UMR INTERTRYP, Parasite Infectiology and Public Health Research group. IRD, CIRAD, Montpellier, France. [13]Laboratory of Parasitology, Micology and Medical Entomology, Istituto Zooprofilattico Sperimentale delle Venezie, Legnaro Padova, Italy. [14]Departamento de Producción y Sanidad Animal, Salud Pública Veterinaria y Ciencia y Tecnología de los Alimentos (PASAPTA), Facultad de Veterinaria, Universidad CEU Cardenal Herrera, Valencia, Spain. [15]Laboratorio de investigación de Entomología, Departamento de Zoología, Facultad de Ciencias Biológicas, Bloque B, Universidad de Valencia, Valencia, Spain. [16]Department of Animal Health, Animal Health and Zoonosis Research Group (GISAZ), UIC Zoonosis and Emerging Diseases (ENZOEM), University of Cordoba, Cordoba, Spain. [17]Department of Animal Production and Health, Veterinary Public Health and Food Science and Technology (PASAPTA), Facultad de Veterinaria, Universidad Cardenal Herrera-CEU, CEU Universities, Valencia, Spain. [18]Applied Zoology and Animal Conservation Group, University of the Balearic Islands (UIB), Palma de Mallorca, Spain. [19]Animal Health Department, The AgriFood Institute of Aragon (IA2), School of Veterinary Medicine, University of Zaragoza, Zaragoza, Spain. [20]Faculty of Science, The University of Melbourne, Parkville, Australia. [21]Department of Veterinary Medicine, University of Bari, Bari, Italy. [22]Faculty of Veterinary Sciences, Bu-Ali Sina University, Hamedan, Iran. [23]Division of Science, Research and Development, Federal Ministry of Defence, Vienna, Austria. [24]Université de Reims Champagne Ardenne, ESCAPE EA7510, USC ANSES VECPAR, SFR Cap Santé, UFR de Pharmacie, Reims, France. [25]Department of Clinical Sciences of Veterinary Medicine, Faculty of Veterinary Medicine, University of Sarajevo, Sarajevo, Bosnia and Herzegovina. [26]Department of Biology, Ecology Section, Faculty of Science, VERG Laboratories, Hacettepe University, Ankara, Turkey. [27]Department of Pathobiology and Epidemiology, Veterinary Faculty, University of Sarajevo, Sarajevo, Bosnia and Herzegovina. [28]University of Pannonia, Sustainability Solutions Research Lab, Veszprém, Hungary. ✉email: attilatrajer@gmail.com; trajer.attila@mk.uni-pannon.hu

