## [Peer Review File · Communications Biology]

Reviewers' comments:

Reviewer #1 (Remarks to the Author):

ABSTRACT BY REVIEWER

In this study, Kniha et al. utilized presence-only records, bioclimatic data and two mitochondrial fragments to investigate the spatial distribution and genetic diversity of *Phlebotomus mascittii* in central Europe. The distribution of *Ph. mascittii* was also predicted based on modeled habitat suitability at seven time periods after the last glaciation event. The authors used models of habitat suitability to predict the distribution of *Ph. mascittii* during seven post-glacial maximum periods after the last glaciation event. The COI and CytB mitochondrial fragments were used to estimate demographic indicators for predicting genetic expansion and reconstructing past demographic events using Bayesian methods. The average nucleotide diversity (π) was low at 0.0008 and 0.0084, for COI and Cytb, respectively, with several Cytb singletons being sequenced. The Bayesian methods, which were calibrated with a general insect substitution rate, estimated the common ancestor of the current *Ph. mascittii* haplotypes to have lived between 26 and 213 thousand years ago. The authors made interesting geographic connections and proposed --although speculative-- two migration routes. However, isolation by distance was not detected in this dataset, suggesting that genetic divergence does not correspond to geographic distance. While the effect of sampling bias on the models is unknown, the authors suggest that the France territory likely provided optimal climatic refugia for the flies after the last glaciation event (~17 ka).

MAJOR CONCERNS:

Additional sampling is needed from western and southern Spain, as well as eastern Europe to enhance the representation of genetic diversity.

Can you please provide additional information regarding the protein alignment? It would be helpful to confirm that conserved regions have minimal non-silent mutations. My primary concern is the possibility of NUMTS amplification, which can be challenging to identify as they may not contain indels or stop codons. To address this issue, it may be advisable to remove any singleton mutations that appear suspicious.

Mitochondrial DNA and nuclear DNA evolve at different rates, and this difference can potentially overestimate divergence times. Can you please provide several references supporting the use of mitochondrial DNA in inferring demographic processes for this taxon?

Line 192: could you please confirm if singletons were excluded from the analyses, and if not, please provide a rationale for their inclusion.

Line 396: could you please provide clarification on why uncorrected distances were utilized.

Line 405: The data suggest that a partition scheme is warranted as the Cytb gene harbors more mutations than COI, resulting in different population structures. Please provide an explanation.

Line 408: Could the authors clarify whether the substitution rate of 0.0115 substitutions per MY was selected based on reference 83? If so, why did the authors decide on using a butterfly parameter for this study. Please consider using the substitution rate range for the target species, as suggested by previous studies (<https://doi.org/10.1093/molbev/mss150>, <https://doi.org/10.1046/j.1365-2583.1997.00175.x>). Additionally, a corrected substitution rate for Psychodidae has been proposed (<https://doi.org/10.1371/journal.pntd.0006614>).

Line 433: Please provide a description the "heterogeneous" methods.

Line 451: Did the authors develop the software used to produce the climatic suitability maps? If so, where was the script deposited? Please clarify.

Reviewer #2 (Remarks to the Author):

The manuscript is written well and has valuable information for international scientific community.

Reviewer #3 (Remarks to the Author):

The study is novel and has a great interest. The authors generally do a great job in this paper. But some notices are appended below concerning small linguistic observations and one scientific claim.

Dear David Favero, PhD, dear reviewers,

*Thank you very much for your valuable comments and for giving us the chance to revise our work. Based on these comments, we have now carefully and substantially revised our manuscript, and we believe that the currently resubmitted version reflects correctly the points raised by the reviewers. Addressing reviewer 1's request for additional sampling, we were able to include new data for the Balkan area (Slovenia and Bosnia), and we provide a detailed explanation of the observed absence of *Phlebotomus mascittii* in southern and south-western Spain, making it unable to include data from this region as none exists (see answer below). In addition, the newly compiled molecular data was included in modelling and network analysis, which were recalculated, and respective passages were edited accordingly in the manuscript.*

*All comments made by the reviewers were fully addressed, as explained in detail below! We hope to now meet your and the reviewers' expectations for publishing in *Communications Biology*.*

Dear Dr Trájer,

First, I would like to apologize for the substantial delay in returning a decision on your manuscript.

Your manuscript entitled "Reconstructing the post-glacial spread of the sand fly *Phlebotomus mascittii* Grassi, 1908 (Diptera: Psychodidae) in Europe" has now been seen by 3 referees, whose comments are appended below. You will see from their comments copied below that while they find your work of potential interest, they have raised quite substantial concerns that must be addressed. In light of these comments, we cannot accept the manuscript for publication, but would be interested in considering a revised version that addresses these serious concerns.

We hope you will find the referees' comments useful as you decide how to proceed. Should further experimental data or analysis allow you to address these criticisms, we would be happy to look at a substantially revised manuscript. However, please bear in mind that we will be reluctant to approach the referees again in the absence of major revisions.

Please address all the concerns of the reviewers in your revised manuscript, including the concern raised by Reviewer #1 that more sampling is needed from western and southern Spain as well as from eastern Europe to enhance the representation of genetic diversity.

We are committed to providing a fair and constructive peer-review process. Do not hesitate to contact us if you wish to discuss the revision or if there are specific requests from the reviewers that you believe are technically impossible or unlikely to yield a meaningful outcome.

If you decide to submit a revised version, we ask that you ensure your manuscript complies with our editorial policies. Please see our revision checklist for guidance on formatting the manuscript and complying with our policies. A comprehensive guide to our formatting requirements for final submissions is also available for your reference here.

Best regards,

David Favero, PhD

Associate Editor

Communications Biology

Referee expertise:

Referee #1: insect ecology, phylogenetics

Referee #2: modeling habitat suitability for insects

Referee #3: species distribution modeling, insect biogeography

Reviewers' comments:

Reviewer #1 (Remarks to the Author):

ABSTRACT BY REVIEWER

In this study, Kniha et al. utilized presence-only records, bioclimatic data, and two mitochondrial fragments to investigate the spatial distribution and genetic diversity of *Phlebotomus mascittii* in central Europe. The distribution of *Ph. mascittii* was also predicted based on modeled habitat suitability at seven time periods after the last glaciation event. The authors used models of habitat suitability to predict the distribution of *Ph. mascittii* during seven post-glacial maximum periods after the last glaciation event. The COI and CytB mitochondrial fragments were used to estimate demographic indicators for predicting genetic expansion and reconstructing past demographic events using Bayesian methods. The average nucleotide diversity (π) was low at 0.0008 and 0.0084 for COI and Cytb, respectively, with several Cytb singletons being sequenced. The Bayesian methods, which were calibrated with a general insect substitution rate, estimated the common ancestor of the current *Ph. mascittii* haplotypes to have lived between 26 and 213 thousand years ago. The authors made interesting geographic connections and proposed, although speculative, two migration routes. However, isolation by distance was not detected in this dataset, suggesting that genetic divergence does not correspond to geographic distance. While the effect of sampling bias on the models is unknown, the authors suggest that the French territory likely provided optimal climatic refugia for the flies after the last glaciation event (~17 ka).

MAJOR CONCERNS:

Additional sampling is needed from western and southern Spain, as well as eastern Europe to enhance the representation of genetic diversity.

While we now present an updated dataset including sequences of ten *Ph. mascittii* specimens originating from Slovenia and Bosnia and Herzegovina, this request needs detailed clarification:

We understand that some countries/regions, e.g. Spain or Balkan seem to be not as well represented as others. However, this is not based on little sampling effort, but on the observed absence of *Phlebotomus mascittii* in these regions confirmed by previous extensive sampling surveys. Here we want to provide a clarification in detail:

Eastern Europe, Balkan: Sand fly surveys have been conducted for decades in Balkan countries; however, the sampling activities were often limited due to the political unrest in the region. Moreover, old records are sometimes unclear as the geographical region is often referred to as Yugoslavia (thus the unclear origin of samples) and DNA is not available. A recent study, published by one of the coauthors and colleagues (Dvorak et al. 2020, <https://doi.org/10.1186/s13071-020-04448-w>), reported historical and recent records of sand flies in eight Balkan countries. Recent extensive data (2014–2016) was collected within the VectorNet project that represented sand fly surveillance activity in Balkan countries after a hiatus of several decades due to the afore mentioned reasons. Within the 8490 sand flies collected at 358 locations in eight Balkan countries, only a mere 12 specimens were *Phlebotomus mascittii* (5 Serbia, 4 Kosovo, and 3 BIH). This demonstrates that despite considerable sampling effort, the species is not often encountered, and additional samples are not readily available.

We used available DNA sequences of *Ph. mascittii* from GenBank in the first version of our manuscript. In this revised version, we also provide new sequences of the three Bosnian specimens and present two additional sequences of specimens newly collected in BIH to further boost our dataset, as suggested by the reviewer. Unfortunately, neither DNA nor sequences of specimens collected in Kosovo are available, despite a new sampling effort in summer 2023. Additionally, a recent survey (2022) in Kosovo by E. Kniha and a colleague from Kosovo trapping at 114 locations in all districts of the country did not reveal any *Ph. mascittii* (Xhekaj et al. 2023).

Considering a few recent records from Serbia, Kosovo, and Bosnia, as well as the observed absence recently reported in several other Balkan countries (North Macedonia, Montenegro, and Croatia), we are very certain that *Ph. mascittii* is very rare in the Balkan region and thus hardly observed. We truly believe that the current data reflects more or less the actual distribution of *Ph. mascittii* in the Balkans. Therefore, we are confident that our modelling effort is not biased by unrepresentative sampling, and we analyzed all available molecular and occurrence data.

Western and Southern Spain: Due to the endemicity of both human and canine leishmaniasis, Spain is certainly one of the most sampled European countries regarding sand flies. One of our coauthors (Bravo-Barriga et al. 2022, doi: 10.3897/zookeys.1106.81432) recently published a checklist and review of sand flies in Spain, taking 138 publications into account. They have gathered data from nearly 20 years of surveillance across 1040 municipalities; thus, geographically and temporally, there has been very extensive sampling. Despite such a wide scope, only a few specimens of *Ph. mascittii* have been detected, specifically in Cantabria (specimens analyzed in this study) and Basque Country (old records, no DNA available). Additionally, studies conducted by other teams throughout Spain in different habitats have also failed to detect its presence. As all European sand flies are trapped with the same methods, *Ph. mascittii* can be noted as absent in the western and southern parts of Spain. We have checked with authors of other studies, and to the best of our knowledge, we have used all existing DNA and sequence data in this study.

Can you please provide additional information regarding the protein alignment? It would be helpful to confirm that conserved regions have minimal non-silent mutations. My primary concern is the possibility of NUMTs amplification, which can be challenging to identify as they may not contain indels or stop codons. To address this issue, it may be advisable to remove any singleton mutations that appear suspicious.

Yes, indeed, in theory, we could have amplified NUMTs instead of the original mitochondrial sequences. We, however, do not think this is the case here for the following reasons: First, there are no indels in our data, and second, no internal stop codons are present as judged from amino acid alignments (we provide them now for both genes as Supplementary Tables 2 and 3). Of course, this is no definitive proof that our sequences are actual mitochondrial sequences and not NUMTs. But if we had sequenced NUMTs, all our sequences should be NUMTs, as it seems unlikely that this intraspecific dataset (with little overall genetic variation) contains mixed data. Mutations do occur in species and populations, and the finding of singletons is not exceptional but rather quite normal in intraspecific datasets like ours. Hence, we refrained from excluding them. Furthermore, we have sequenced both genes in both directions to verify potentially ambiguous positions. So, we are confident that our data is OK and does not contain NUMTs.

Mitochondrial DNA and nuclear DNA evolve at different rates, and this difference can potentially overestimate divergence times. Can you please provide several references supporting the use of mitochondrial DNA in inferring demographic processes for this taxon?

Yes, the reviewer is right; mitochondrial and nuclear DNA evolve at quite different rates. Typically, mitochondrial DNA has a much higher substitution rate than nuclear DNA. Therefore, mitochondrial DNA sequences are much better suited for resolving recent events than nuclear sequence data (unless this difference is compensated for by sequencing a large number of nuclear loci) and have been the marker of choice in many studies inferring demographic processes (and other population genetic and phylogeographic patterns) not only in sandflies but in animals in general.

Please see the following literature:

- Essegir, S., Ready, P. D., Killick-Kendrick, R., & Ben-Ismaïl, R. (1997). Mitochondrial haplotypes and phylogeography of *Phlebotomus* vectors of *Leishmania major*. *Insect molecular biology*, 6(3), 211-225. <https://doi.org/10.1046/j.1365-2583.1997.00175.x>
- Mahamdallie, S. S., Pesson, B., & Ready, P. D. (2011). Multiple genetic divergences and population expansions of a Mediterranean sandfly, *Phlebotomus ariasi*, in Europe during the

Pleistocene glacial cycles. *Heredity*, 106(5), 714-726.

<https://www.nature.com/articles/hdy2010111>

- Kasap, O. E., Dvorak, V., Depaquit, J., Alten, B., Votycka, J., & Volf, P. (2015). Phylogeography of the subgenus *Transphlebotomus* Artemiev with description of two new species, *Phlebotomus anatolicus* n. sp. and *Phlebotomus killicki* n. sp. *Infection, Genetics and Evolution*, 34, 467-479. <https://doi.org/10.1016/j.meegid.2015.05.025>
- Pech-May, A., Ramsey, J. M., Gonzalez Ittig, R. E., Giuliani, M., Berrozpe, P., Quintana, M. G., & Salomon, O. D. (2018). Genetic diversity, phylogeography and molecular clock of the *Lutzomyia longipalpis* complex (Diptera: Psychodidae). *PLoS neglected tropical diseases*, 12(7), e0006614. <https://doi.org/10.1371/journal.pntd.0006614>
- Pavlou, C., Dokianakis, E., Tsigotakis, N., Christodoulou, V., Özbek, Y., Antoniou, M., & Poulakakis, N. (2022). A molecular phylogeny and phylogeography of Greek Aegean Island sand flies of the genus *Phlebotomus* (Diptera: Psychodidae). *Arthropod Systematics & Phylogeny*, 80, 137-154. <https://doi.org/10.3897/asp.80.e78315>
- Lozano-Sardaneta, Y. N., Díaz-Cruz, J. A., Viveros-Santos, V., Ibáñez-Bernal, S., Huerta, H., Marina, C. F., ... & Becker, I. (2023). Phylogenetic relations among Mexican phlebotomine sand flies (Diptera: Psychodidae) and their divergence time estimation. *Plos one*, 18(6), e0287853. <https://doi.org/10.1371/journal.pone.0287853>

Line 192: could you please confirm if singletons were excluded from the analyses, and if not, please provide a rationale for their inclusion.

No, singletons were not excluded for the reasons mentioned above. Briefly, we are confident that these are neither sequencing errors nor NUMTs.

Line 396: could you please provide clarification on why uncorrected distances were utilized.

Since we are not interested in inferring phylogenetic trees (where multiple hits might be problematic) but rather want to describe the data, presenting uncorrected p-distances is the method of choice here, also for sakes of comparability with other data or studies.

Line 405: The data suggest that a partition scheme is warranted as the *Cytb* gene harbors more mutations than COI, resulting in different population structures. Please provide an explanation.

Here, we respectfully disagree with the reviewer. Yes, normally, we would opt for partitioning the data by gene, or if suggested by an objective model-testing approach like, e.g., partitionfinder, by codon position (either per gene or for the whole dataset). However, because of the very little variation we see per gene, data partitioning would result in overparameterization, resulting in non-convergence of the BSP analyses. Indeed, we did some preliminary BSP runs with partitioned data, and these did not converge. Please also note that since COI and *cytb* are mitochondrial genes, these are linked (no recombination in the mitochondrial genome), and hence they should not give different results in population structure. Indeed, the results of COI and *cytb* are consistent, with one haplotype shared by the majority of samples and samples from Corsica somewhat separated. The presence of a few singletons and other low-frequency haplotypes is perfectly normal and expected for this kind of dataset.

Line 408: Could the authors clarify whether the substitution rate of 0.0115 substitutions per MY was selected based on reference 83? If so, why did the authors decide on using a butterfly parameter for this study. Please consider using the substitution rate range for the target species, as suggested by previous studies (<https://doi.org/10.1093/molbev/mss150>, <https://doi.org/10.1046/j.1365-2583.1997.00175.x>). Additionally, a corrected substitution rate for Psychodidae has been proposed (<https://doi.org/10.1371/journal.pntd.0006614>).

Yes, the substitution rate used was the general mitochondrial substitution rate for insects proposed by reference 83. And yes, the reviewer is right that there is quite some variation around this general estimate and that it differs slightly among mitochondrial genes and taxa. Nonetheless, this rate is being used in the vast majority of studies that include divergence time estimates or infer population

size changes in insects. No substitution rate estimate exists for COI and CYTB in Phlebotomus. Also, the studies suggested by the reviewer are of no particular use for this purpose:

1. Obbard et al. (2012) (<https://doi.org/10.1093/molbev/mss150>) employ a nuclear multilocus sequence dataset to study the divergence of Drosophila. Neither COI nor CYTB are included in their dataset.
2. Essegir et al. 1997 (<https://doi.org/10.1046/j.1365-2583.1997.00175.x>) used a fragment spanning part of the cytb and part of the ND4 gene. Hence, though also mitochondrial, it's not exactly the same fragment(s) we sequenced and analyzed.
3. Pech-May et al. (2018) (<https://doi.org/10.1371/journal.pntd.0006614>) used the rate of Essegir and corrected it (for time dependency of the molecular clock) following Ho et al. (2005) Mol Biol Evol. However, they did not specify in detail how exactly they corrected the rate or which of the various corrections listed by Ho et al. (2005) (for different potentially occurring errors) they employed.

Consequently, to be conservative, we inferred population size changes assuming a minimum and maximum substitution rate of 1.0% and 2.5% per million years, respectively, which covers the range typically employed or inferred in studies on insects. Note that the rate inferred by Essegir et al. (1997), though not exactly from the same gene fragments, falls in this very same range.

Line 433: Please provide a description the "heterogeneous" methods.

Apologies; we were unclear. The term heterogenous methods refers to the below-presented acquisition of modeling data. We have rephrased the sentence, and it should be clear now.

Line 451: Did the authors develop the software used to produce the climatic suitability maps? If so, where was the script deposited? Please clarify.

The algorithms were run in the raster calculator of Quantum GIS.

It should be noted that the temperature-like climatic variables (from bio1 to bio11) are multiplied by 10 in the original climatic database, so these values appear in the same way in the equation.

The script of the utilized equation was as follows (we also provide it as Supplementary Text 1 for the readers):

```
("bio_1@1" >= 79 ) + ("bio_1@1" <= 167 ) + ("bio_2@1" >= 37 ) + ("bio_2@1" <=88 ) + ("bio_3@1" >= 19 ) + ("bio_3@1" <= 33 ) + ("bio_4@1" >= 4620 ) + ("bio_4@1" <= 7785 ) + ("bio_5@1" >= 214 ) + ("bio_5@1" <= 302 ) + ("bio_6@1" >= -52 ) + ("bio_6@1" <= 82 ) + ("bio_7@1" >= 176 ) + ("bio_7@1" <= 308 ) + ("bio_8@1" >= 34 ) + ("bio_8@1" <= 207 ) + ("bio_9@1" >= -5 ) + ("bio_9@1" <= 248 ) + ("bio_10@1" >= 172 ) + ("bio_10@1" <= 258 ) + ("bio_11@1" >= -10 ) + ("bio_11@1" <= 99 ) + ("bio_12@1" >= 551 ) + ("bio_12@1" <= 1846 ) + ("bio_13@1" >= 65 ) + ("bio_13@1" <= 219 ) + ("bio_14@1" >= 10 ) + ("bio_14@1" <= 103 ) + ("bio_15@1" >= 10 ) + ("bio_15@1" <= 53 ) + ("bio_16@1" >= 195 ) + ("bio_16@1" <= 641 ) + ("bio_17@1" >= 39 ) + ("bio_17@1" <= 324 ) + ("bio_18@1" >= 49 ) + ("bio_18@1" <= 590 ) + ("bio_19@1" >= 99 ) + ("bio_19@1" <= 406 )
```

Reviewer #2 (Remarks to the Author):

Brief summary of the manuscript

The manuscript is well written and have sound information regarding the prediction of habitat suitability analysis of sand fly in Europe. The Authors have implemented multiple approaches of modeling and study was designed well. Results of the study have been presented well and would attract a number of international readers.

Overall impression of the work

The work is very nice and have significant information.

Thank you very much. Please find our comments below.

I have some miner suggestions for the improvement of manuscript.

Abstract:

It may be improved by adding some more methodology and future suggestions on the gaps of the study.

We added and edited sentences in the abstract on M&M and provided future suggestions.

Results

Line 173-203 looks like the part of materials and methods. They may be adjusted accordingly.

We agree; we have integrated the first passage in M&M accordingly in the section "Sequence editing and alignment."

Reviewer #3 (Remarks to the Author):

The study is novel and has a great interest. The authors generally do a great job in this paper. But some notices are appended below concerning small linguistic observations and one scientific claim.

This study aimed to illustrate the effect of climatic and sea level changes on the expansion of *Ph. mascittii* from its glacial refugia using an integrated approach linking phylogeographic data, climate modelling, and network analysis to unlock new insights into the process of post-glacial sand fly colonization in Europe. The study is novel and has a great interest. The authors generally do a great job in this paper. But some notices are appended below concerning small linguistic observations and one scientific claim.

Thank you very much, we made sure to address all comments:

1. Linguistic observations:

Line 91: The first one was through.....

Done!

Line 94: and the post-glacial...

Done.

Line 142: Central and Eastern Europe were...

Done!

Line 143:a considerable decrease in pre-glacial...

Thank you, done!

Line 208: the Bay of Biscay and the Gulf....

Done!

Line 268: Kapolcs Valley....

Absolutely!

Line 294: potentials....

Respectfully, potential seems correct to us, as potential refers to the dispersal routes!

Line 324: different flight abilities.....

Changed!

Line 364: was....

Yes!

Line 438: the center coordinates....

Done!

Line 475: represents.....

Yes!

2. Minor Claim:

In the Methods section, the authors didn't illustrate the source of the 13 bioclimatic data. Also, why they used only 13 out of 19 covariates? Moreover, it is well known that bioclimatic layers 8-9 and 18-19 have spatial artifacts and could be omitted from predictive models.

In the revised version of the models, all 19 variables were applied. Anyway, we checked the mentioned variables, and notable spatial pattern bias can be observed in the following regions: the Amazonian part of South America; Central, East, and South Africa; the Indian subcontinent; Central America; and North Australia. In the case of Europe, which was in the scope of the study, notable problems in the basic bioclimatic models cannot be seen. The validity of this statement can be checked by downloading the tiff files from the PaleoClim database* and depicting them using a GIS.

*: <http://www.paleoclim.org>

On the other hand, to reduce the multicollinearity among bioclimatic variables, the Pearson correlation coefficient is used to hinder the correlation among the variables and the study neglected this issue!!!

The used climate envelope-like modelling technique does not require testing for collinearity. It is due to the fact that this modelling technique considers the factors independently from each other based on a similar concept to the barrel analogy of Liebig's law of the minimum. A related text to this important question was added to the MS.

Furthermore, it isn't clear to the readers or me as a reviewer what is the modelling program used to develop the resulting models in Figures 4 and 5!!!

Quantum Gis was used to create the climatic suitability models. This fact was now clearly declared in the revised version of the study in a relevant part of the methodology.

These raised issues in minor claims must be illustrated. But generally, the study is very interesting!!!

Decision: Accept after revision

REVIEWERS' COMMENTS:

Reviewer #1 (Remarks to the Author):

This reviewer would like to congratulate the authors for their hard work. All my comments have been thoroughly addressed in their replies.

Reviewer #3 (Remarks to the Author):

The authors do a great job in the revision process!

Subject: Response to referees

Veszprém, November 15th 2023

Dear reviewers,

Thank you very much for the very productive revision process and for all the valuable comments on our work with the title:

“Reconstructing the post-glacial spread of the sand fly *Phlebotomus mascittii* Grassi, 1908 (Diptera: Psychodidae) in Europe”

All raised points were already addressed in the previous revision process!

On behalf of all coauthors,

Yours sincerely,
Attila J. Trájer MD PhD